# Effects of Dried Tea Residues of Different Processing Techniques on the Nutritional Parameters, Fermentation Quality, and Bacterial Structure of Silaged Alfalfa

**DOI:** 10.3390/microorganisms12050889

**Published:** 2024-04-29

**Authors:** Xingcheng Lei, Binbin Na, Tong Zhou, Yuangan Qian, Yixiao Xie, Yulong Zheng, Qiming Cheng, Ping Li, Chao Chen, Hong Sun

**Affiliations:** College of Animal Science, Guizhou University, Guiyang 550025, China; leixc9972@163.com (X.L.); 15969457062@163.com (B.N.); zlilit@126.com (T.Z.); qianyg183@163.com (Y.Q.); xieyx@gzu.edu.cn (Y.X.); ylzheng3@gzu.edu.cn (Y.Z.); qmcheng@gzu.edu.cn (Q.C.); lip@gzu.edu.cn (P.L.); gzgyxgc3855218@163.com (C.C.)

**Keywords:** alfalfa silage, tea residues, nitrogen components degradation, antioxidant activity

## Abstract

The effects of dried tea residues on the nutritional parameters and fermentation quality, microbial community, and in vitro digestibility of alfalfa silage were investigated. In this study, dried tea residues generated from five different processing techniques (green tea, G; black tea, B; white tea, W; Pu’er raw tea, Z; Pu’er ripe tea, D) were added at two addition levels (5% and 10% fresh weight (FW)) to alfalfa and fermented for 90 days. The results showed that the tea residues increased the crude protein (CP) content (Z10: 23.85%), true protein nitrogen (TPN) content, DPPH, and ABST radical scavenging capacity, total antioxidant capacity (T-AOC), and in vitro dry matter digestibility (IVDMD) of the alfalfa silage. Moreover, the pH, ammonia-N (NH_3_-N) content, and acetic acid (AA) content decreased (*p* < 0.05). The effects of tea residues were promoted on these indicators with increasing tea residue addition. In addition, this study revealed that the influence of dried tea residues on the nutritional quality of alfalfa silage was greater than that on fermentation quality. Based on the nutrient composition, the addition of B or G to alfalfa silage can improve its silage quality, and these tea byproducts have the potential to be used as silage additives.

## 1. Introduction

With the prevalence of tea beverages, the demand for tea production has increased. As the country of origin of tea, China has the largest tea planting area and highest tea production. In 2022, its tea plantation area reached to 3.39 million hectares and tea production has reached 3.34 million tons [1]. With the increase in the demand for tea products and tea drinks, large amounts of tea residues are produced. Tea residue refers to the solid organic waste generated during the production, processing, sale, and tea beverage production process of tea. During the processes of tea picking and processing, a large number of byproducts called dry tea residue, such as coarse tea, broken tea, and aged tea, are produced. After making tea drinks, a large amount of wet tea residue is produced [2,3]. According to relevant reports, it was estimated that approximately 190,400 tons of tea residue were produced in India every year, which accounted for 22.2% of the annual tea production [3]. As the major tea producing and consuming country, China has more amounts of tea residue and is exploring how to effectively utilize tea residue [4].

Tea is so favored because of its unique nutrient and active substance profiles [5,6,7,8]. Studies have shown that tea residues contain a variety of valuable nutrients; for example, tea protein has a good hypoglycemic and hypolipidemic effect, and the polyphenols in tea play a significant role in promoting human health [5]. Additionally, the dietary fiber in tea has the ability to regulate intestinal health [6]. It also has the ability to promote caloric intake, reduce body fat, improve insulin sensitivity, and reduce the risk of Alzheimer’s disease and other neurodegenerative diseases [7,8]. Bose et al. [9] and Lu et al. [10] reported that green tea polyphenol supplementation in rodent diets suppressed body weight and body fat accumulation induced by a liver fat diet. In addition, tea contains a variety of antioxidant substances, such as catechins, theaflavins, thearubigins, oxidized aromatic acids, flavonoids, gallic acid, and their derivatives [11,12], that can increase the antioxidant capacity of an organism. Moreover, tea also contains more than 60 volatile components, including alcohols, carbonyls, esters, acids, and cyclic compounds [13], and theanine [14]. These characteristic substances make tea one of the most antioxidant plants [11].

The reuse of tea residue waste, which is rich in the abovementioned characteristic substances, is necessary to reduce the waste of valuable compounds. In the research and utilization of tea residues, some studies have focused on the use of tea residues as acoustic materials [15,16], and others have focused on the use of tea as a medical reagent [17,18]. Additionally, there are studies on the use of tea as feed or as an additive in livestock feed. Tea residues, when serve as an ingredient in total mixed rations, can effectively improve ration quality and feeding effects [19,20,21]. Tea addition can improve the microbial composition of pig intestinal tracts and the accumulation of abdominal fat in broilers and can positively affect the quality of egg yolks and the antioxidant capacity of bovine plasma [22,23]. Studies have shown that replacing the diet concentrate in the 20% DM used to feed Holstein cattle with green tea waste can increase the antioxidant activity and vitamin E concentration of bovine plasma without negatively affecting rumen fermentation [24]. In addition, the degradation rate of the fermented tea residue treatment group in the rumen was greater than that of the unfermented tea residue treatment group in the mixed silage of wheat bran and tea residue [25]. In silage development, tea residues can be used as raw materials for silage alone [26] or as additives to silage [27,28,29,30]. However, most studies have focused on wet tea residue, and few studies have focused on dry tea residue as a feed silage additive. Moreover, the content of nutrients and characteristic substances varies among the tea residues produced via different processing techniques [5].

To clarify the effect of dry tea residue additives on different silage parameters, five common dry tea residues with different processing techniques were selected as additives, and alfalfa was used as the raw material. The changes in the nutrition, fermentation parameters, fermentation quality, and microbial community of alfalfa after 90 days of silage were investigated. This paper provides theoretical support and guidance for the future utilization of dry tea residue waste in silage.

## 2. Materials and Methods

### 2.1. Materials and Silage Preparation

Alfalfa (*Medicago sativa* L.) at the full bloom stage was harvested through cutting at a location approximately 5 cm above the ground on 25 September 2021 from an experimental field in Anshun (Guizhou, China) and returned to the laboratory at Guizhou University (Guiyang, China). The alfalfa was manually cut into small pieces with a length of approximately 2~3 cm by a sickle and randomly divided into 11 portions of similar weight. Five dried tea residues were purchased from Guizhou Dachacang Development Co., Ltd. (Guiyang, China), including green tea residues (G), black tea residues (B), white tea residues (W), Pu’er raw tea residues (Z), and Pu’er ripe tea residues (D). These tea residues were randomly added to alfalfa at proportions of 5% and 10% of the fresh weight of alfalfa. There were 11 treatments, namely, CK, B5, B10, D5, D10, G5, G10, W5, W10, Z5, and Z10 (where the letter represents the tea residue and the number represents the percent addition), with 4 replicates per group. Forty-four (11 treatments * 4 replicates) polyethylene vacuum bags were used to silage alfalfa, and the bags were sealed with a vacuum sealer; after 90 days, the fermentation quality, chemical composition, and microbial community of the alfalfa were analyzed.

### 2.2. Fermentation Quality and Chemical Composition Analyses

At the end of ensiling, we removed 10 g from each bag, placed it in a labeled polyethylene bag, and then added 90 mL of sterile 0.85% NaCl solution to each bag. The bags containing the sample and saline were placed in a 4 °C freezer overnight and then filtered with 4 layers of medical gauze, and the resulting filtrate was the extract.

The pH of each extract was determined using a pH meter (pH21I precision type), and the organic acid concentrations {including lactic acid (LA), acetic acid (AA), propionic acid (PA) and butyric acid (BA)} in the extract were determined using high-performance liquid chromatography (HPLC). In addition, the extracts were taken out 40 mL, and we added 10 mL of 25% trichloroacetic acid (TCA) to the extracts. Then, we stored them overnight at 4 °C. Then, the true protein nitrogen (TPN) and non-protein nitrogen (NPN = TN − TPN) concentrations were determined according to the methods of the Association of Official Analytical Chemists [31]. The concentrations of ammonia-N (NH_3_-N) and free amino acid nitrogen (FAA-N) in each extract were determined using the method of Broderick et al. [32], and the peptide nitrogen (PN = NPN − NH_3_-N − FAA-N) concentration was calculated, finally. Furthermore, a small amount of the extract was properly diluted in a sterile environment (dilution factor:10^3^, 10^5^, 10^7^), and after it was added to the medium for 48 h and cultured, the number of major bacteria was recorded.

Secondly, the pre-ensiled alfalfa and silage samples were taken out 150~200 g and placed in a 65 °C oven to dry to constant weight, the DM content was determined, the dried samples were crushed with a high-speed crusher (FW100, Tester Instrument Co., Ltd. Xiamen, China), and the samples were subsequently passed through a sieve with a pore size of 1 mm. The concentrations of curde protein (CP) were analyzed using a Kjeldahl nitrogen analyzer (K9860 Auto-analyser, Jinan Hanon Instruments Co., Ltd., Jinan, China) according to the methods of the Association of Official Analytical Chemists [31]. The water-soluble carbohydrate (WSC) concentration was determined using a spectrophotometer (SP-1920, Shanghai Spectral Instrument Co., Ltd., Shanghai, China), according to the sulfuric acid–anthrone colorimetric method [33]. The concentrations of neutral detergent fiber (NDF) and acid detergent fiber (ADF) were measured using a fiber analyzer (ANKOM^DELTA^, ANKOM Technology, New York, NY, USA) according to the method of Van Soest et al. [34]. The antioxidant activity of the samples was determined by using a Solarbio ABTS radical scavenging ability detection kit, total antioxidant capacity (T-AOC) detection kit, and DPPH free radical scavenging capacity detection kit (Beijing Solarbio Science & Technology Co., Ltd., Beijing, China).

### 2.3. Microbial Diversity Analysis

#### 2.3.1. Extraction of DNA

Ten grams of sample was mixed with 90 mL of sterile 0.85% NaCl solution with shaking for 2 h at 120 r/min in an incubator. The mixture was filtered through four layers of cheesecloth, and the filtrate was centrifuged at 4 °C, 10,000 r/m for 15 min. The deposit was resuspended in 1 mL of sterile 0.85% NaCl solution, and the microbial pellets were obtained through centrifugation at 12,000 r/m and 4 °C for 10 min. Total DNA was extracted using the Power Soil DNA Isolation Kit (MO Bio Laboratories, Inc., Carlsbad, CA, USA).

#### 2.3.2. PCR Amplification and Sequencing

The extracted DNA was sent to Beijing Biomarker Technology Co., Ltd. (Beijing, China) and PCR amplification and bioinformatics analysis were performed on the extracted DNA of all samples. Using a forward primer (50-ACTCCTACGGGAGGCA GCA-30) and a reverse primer (50-GGACTACHVGGGTWTCTAAT-30), the V3–V4 region of 16S rDNA was amplified. The operational taxonomic units (OTUs) were grouped together using the QIIME (v1.8.0) software with a 97% similarity rate. Utilizing the MOTHUR software (v.1.41.1), alpha diversity was calculated and included the OTU, Shannon, Chao1, Ace, and coverage. Classification was conducted at the phylum and genus levels according to the Silva database. Relative abundance comparison and Spearman correlation were used for bacterial community analysis.

### 2.4. In Vitro Incubation and Degradability Measurements

Rumen fluid was collected from cattle (Angus) within 30 min of slaughter, filtered through four layers of gauze, placed in thermoses, maintained anaerobically at 39 °C, and subsequently transported to the laboratory. Artificial rumen buffer was prepared according to the method of Menke et al. [35], made by mixing buffer with the collected rumen fluid in a 3:1 ratio to form an artificial rumen fermentation mixture, and anaerobic conditions were maintained during the preparation process. Samples weighing approximately 0.5000 g were crushed and sieved through a 40-mesh sieve, packed into filter bags (ANKON F57, ANKOM Technology, New York, NY, USA), and randomly divided into 9 groups of 4 bags each for a total of 36 bags (including 3 blanks and 3 replicates per treatment). The samples were placed in 250 mL fermentation bottles by group, and 200 mL of artificial rumen mixed fermentation solution was added under anaerobic conditions. The nine fermentation bottles with samples and mixed fermentation broth were put into a constant-temperature shaking incubator at 39 °C for 72 h. The rotational speed was 110 r/min. The filter bags were removed from the fermentation bottles, rinsed with running water until clarified and colorless, and then placed in an oven at 65 °C to dry until reaching constant weight, after which the degradation rate of the dry matter was calculated.

### 2.5. Statistical Analysis

The data from this experiment were recorded in Excel. Two-way ANOVA was performed in IBM SPSS Statistics version 22 to calculate the fixed effects of the 5 tea residues, 2 addition ratios, and interactions between these 2 treatment parameters. Multiple comparisons were made using Duncan’s test to assess the difference between the data (means). When *p* < 0.05, the difference was considered significant. When *p* < 0.01, the difference was considered extremely significant. All graphics were generated using GraphPad Prism version 9.0 and Origin Pro 2022.

## 3. Results

### 3.1. Characteristics of Fresh Alfalfa before Silage

The chemical components of alfalfa and the five dried tea residues are shown in Table 1. The DM content of alfalfa was 41.62%, and the CP, WSC, NPN, NDF, and ADF contents accounted for 22.38%, 4.00%, 1.01%, 31.83% and 18.36%, respectively, of the DM. The CP, WSC, NPN, NDF and ADF contents were 33.82%, 4.18%, 9.06%, 33.6% and 8.13%, respectively, in G; 29.77%, 7.80%, 7.65%, 33.79% and 14.03%, respectively, in B; 14.47%, 8.49%, 4.05%, 33.93% and 18.55%, respectively, in W; 27.17%, 4.47%, 8.17%, 26.04% and 7.74%, respectively, in Z; and 30.90%, 0.44%, 6.51%, 44.83% and 36.84%, respectively, in D.

### 3.2. Fermentation Quality and Nutritional Quality of Alfalfa Silage in the Control Group

The chemical properties and microbial populations of alfalfa in the control group after 90 days of silage are shown in Table 2. The CP content was 21.49% DM, which was not much different from the content before ensiling. The WSC content was only 1.24% DM, which was nearly 80% lower than the content before ensiling. The NDF and ADF contents were 21.57% and 10.52%, respectively. The pH value was 6.00. The TPN content was 347.95 g/kg TN, and the NPN content was 652.05 g/kg TN (the NH_3_-N, FAA-N, and PN contents accounted for 5.27%, 13.93% and 80.80%, respectively, of the NPN). The contents of LA, AA, and PA were 4.26%, 2.84% and 1.51% of the DM, respectively, and BA was not detected.

### 3.3. Nutritional Quality and In Vitro Dry Matter Digestibility of Alfalfa Silage

As shown in Table 3, the addition of different dried tea residues from different processes and different addition amounts had significant effects on the DM, CP, WSC, ADF, and NDF contents in the alfalfa silage, and their interaction also had a significant effect on these parameters except for the DM content (*p* < 0.05).

The DM content in the treatment groups was significantly greater than that of the CK group (*p* < 0.05), and the effect increased with increasing additive proportion. Moreover, the highest DM contents were observed in G10 (44% FW) and B10 (43% FW). The changes in the CP content in the other groups were similar to the changes in the DM content, except for the groups with added W. Among these groups, the CP content of the D10 group was significantly greater than that of the other groups (*p* < 0.05). The WSC contents of the B5, B10, G10, and W10 groups were greater than that of the CK group, and that of the other groups was lower than that of the CK group. With different tea-residue addition proportions and processing techniques, the NDF and ADF contents of the alfalfa silage were greater than those in the CK group, except for those in the G5 and Z5 groups.

The tea residues prepared via five different processing techniques (G, B, W, Z, and D) increased the IVDMD of alfalfa, but the addition of D did not significantly increase the IVDMD of alfalfa silage (Figure 1, *p* > 0.05). In contrast to the effect of the addition of other tea residues, the increase in the IVDMD of alfalfa caused by the addition of G decreased with an increasing proportion. The above results showed that the addition of tea residue could increase the IVDMD of alfalfa silage, which has a certain reference value for livestock feeding activities.

### 3.4. Effects of Tea Residues on the Nitrogen Composition of Alfalfa Silage

The effects of different dried tea residues and proportions on the TPN, NPN, PN, NH_3_-N, and FAA-N contents in alfalfa silage are shown in Table 4. The different processing techniques and proportions of tea residue had significant effects on the content of nitrogen components in the alfalfa ensile (*p* < 0.05). Among them, the type of dried tea residue and additive proportion had a significant interaction affect in the contents of TPN, NPN, and PN after ensiling. Compared with that in the CK group, the TPN content in the alfalfa ensiled with tea residues significantly increased, and it increased with increasing tea residue proportion. Moreover, the TPN content in the D10 group was the highest, at 507.34 g/kg TN. Compared with TPN content, the NPN and PN contents showed the opposite trend; that is, the NPN and PN contents in the tea residue groups were significantly lower than those in the CK group (*p* < 0.05), and the difference increased with an increasing tea residue proportion. The lowest contents of NPN and PN were observed in the D10 group and were 492.7 g/kg TN and 403.4 g/kg TN, respectively.

The NH_3_-N contents of alfalfa silage in the tea residue-treated groups were also lower than that in the CK group, and the differences increased with an increasing tea residue proportion. Specifically, the NH_3_-N contents in the tea residues groups were significantly lower than that in the CK group, except for in the D5, D10, W5 and Z5 groups (*p* < 0.05). The lowest content of NH_3_-N was observed in the B5 group (NH_3_-N=17.1 g/kg TN). However, there were no significant interactions between dried tea residue type and proportion (*p* < 0.05). The FAA-N content in the groups with added tea residue was lower than that in the CK group, except for that in the B5 group. The FAA-N content of D10 significantly differed from that of the CK group, at 59.4 g/kg TN (*p* < 0.05). In addition, there was no significant interaction effect between the two factors on the FAA-N content (*p* > 0.05).

### 3.5. Effects of Tea Residues on the Fermentation Characteristic of Alfalfa Silage

As shown in Table 5, the addition of dried tea residues prepared using different processing processes had a significant effect on the pH, LA, AA, and PA contents of the ensiled alfalfa while the proportion of dried tea residue had a significant effect only on the pH and LA content of the ensiled alfalfa (*p* < 0.05). Moreover, different proportions of tea residues prepared using different processing techniques had significantly interactional effects on the LA and AA contents of alfalfa after ensiling (*p* < 0.05). The pH values of the G5, B10, and Z10 group were significantly lower than that of the CK group (*p* < 0.05), and the pH of the B10 group was the lowest, at 5.71. Compared with those in the CK group, the LA and AA contents of alfalfa silage in the tea residue groups were significantly lower (*p* < 0.05). The PA content of alfalfa in the tea residue-treated groups was lower than that in the CK group, except for that in the G10 and B10 groups. The PA contents of the D5 and D10 groups were significantly lower than that of the CK group (*p* < 0.05). At the same time, BA was detected in the group with D while BA was not detected in the other groups.

### 3.6. Effects of Tea Residues on the Antioxidant Property of Alfalfa Silage

The overall changes in the three tested antioxidant indicators were basically the same; that is, the addition of four tea residues (G, B, W, and Z) significantly increased the DPPH and ABTS free radical scavenging capacities and the T-AOC of alfalfa while the addition of D significantly reduced these three indicators (*p* < 0.05). The DPPH radical scavenging capacity of alfalfa ensiled with G was the highest, at 89.2%, and its scavenging ability was approximately twice that of the CK group (37.5%, Figure 2A). There was no significant difference between the two added proportions of the four tea residues (G, B, W, and Z) for the ABTS radical scavenging capacity of the ensiled alfalfa (*p* < 0.05, Figure 2B). With the addition of different tea residues, the T-OAC of alfalfa differed, with overall significant increases with an increasing tea residue proportion except for the group with D (Figure 2C).

### 3.7. Effects of Tea Residues on the Bacterial Diversity Structure of Alfalfa Silage

#### 3.7.1. Bacterial Alpha Diversity Index Analysis

The effects of different dried tea residues and different proportions on the bacterial alpha diversity of alfalfa ensiled for 90 days are shown in Table 6. The coverage index was greater than 99% for all silage samples, indicating sufficient sequencing depth coverage such that most bacteria had been identified. In general, the number of OTUs and the Shannon, Simpson, Chao 1, and Ace indices were lower in alfalfa with added tea residues than in the control, while the number of OTUs and the value of each index decreased with an increasing addition proportion, except for the B5 group. The values of the OTUs and other indices were lowest detected in the W10 group.

#### 3.7.2. Analysis of Bacterial Community Structure Based on Phylum and Genus Level

The effects of different dried tea residues and different proportions on bacterial communities of alfalfa at the phylum and genus levels are shown in Figure 2A,B. After 90 days of ensiling, the main species detected in alfalfa were Firmicutes, Cyanobacteria, and Proteobacteria. There was a difference in the relative abundance of microorganisms at the phylum level between these treatment groups and the CK group. The addition of tea residues reduced the relative abundance of Proteobacteria in alfalfa, and the relative abundance in the G5 group was the lowest. The relative abundance of Cyanobacteria in the W10 group was significantly greater than that in the other treatment groups with added tea residue (*p* < 0.05). Overall, regardless of the changes in the relative abundance of various bacteria, the relative abundance of Firmicutes in all silage samples was greatest, and Firmicutes was still the dominant population of bacterial communities in alfalfa.

At the genus level, the bacterial species detected were *Weissella, Lactobacillus, Enterococcus, Bacillus, Enterobacter, Kosakonia, Pantoea, Lactococcus, Pediococcus, Clostridium (Clostridium_sensu_stricto_12)*, and other bacteria not classified elsewhere. The results showed that the addition of different proportions of tea residues had different effects on the microbial structure of alfalfa at the genus level. The relative abundance of *Pantoea* in the bacterial community in all the ensiling samples was greater than that in the control, but the difference was significant only in the W10 group. In addition, the relative abundance of *Bacillus*, which became the main dominant bacteria, in the D treatment group was significantly greater than that in the other treatment groups (*p* < 0.05). Moreover, the relative abundance of *Pediococcus* in the Z10 treatment group was also significantly greater than that in the other groups, including the CK group (*p* < 0.05).

In terms of the types of tea residue added to alfalfa, the relative abundance of *Weissella* in alfalfa in the G, B, W, and Z treatments was greater than that in the CK group. Among them, the relative abundance of *Weissia* was greatest in the B10 group, accounting for 32% of the total bacterial richness. In addition, the relative abundance of *Pediococcus* in the Z10 group was greater than that in the other groups. There were obvious differences between groups with different proportions of the same kind of tea residue, but the change was no regularity. This indicated that the proportion of tea residue added had a great influence on the microbial community structure of the alfalfa silage. However, regardless of how the addition of tea residue affects the bacterial structure of silage samples, the dominant bacteria at the genus level were *Lactobacillus, Enterococcus,* and others type bacterial genera.

#### 3.7.3. Spearman Correlation Analysis of Chemical Compositions and Bacterial Communities of Alfalfa

The Spearman correlation heatmap of the bacterial communities and chemical components of alfalfa ensiled with different proportions of different tea residues is shown in Figure 3C. There was a significant negative correlation between *Lactiplantibacillus* abundance and the DM content (R = −0.35), between *Enterococcus* abundance and the PA content (R = −0.408), and between *Bacillus* abundance and the CP (R = −0.49) and WSC (R = −0.38) contents of alfalfa silage (*p* < 0.05). There were significant positive correlations between *Bacillus* abundance and the NPN content (R = 0.45), the NH_3_-N content (R = 0.41), and the PN content (R = 0.42).There were significant positive correlations between *Kosakonia* abundance and the NDF content (R = 0.35), PA content (R = 0.35; *p* < 0.05). *Lactococcus* abundance was also significantly positively correlated with these two indices (NDF content, R = 0.40; PA content, R = 0.36; *p* < 0.05).

## 4. Discussion

Ensiling is a common way to preserve moist forage crops and can prolong storage time and improve feed palatability via LA fermentation under anaerobic conditions [36,37]. The ideal silage fermentation conditions of a crop are based on a low pH with little or no BA while maintaining a high LA content [36,38,39,40], because lactic acid bacteria (LAB) metabolize WSCs in silage to produce organic acids dominated by LA and AA under anaerobic conditions.

In this study, the DM content of alfalfa before and after silage was in the range of 36~42%, which was slightly greater than the DM content in the study by Su et al. [41] and Yang et al. [42] but basically consistent with Li et al. [43]. In addition, the DM content further increased after the addition of tea residues. This may be due to the high DM content of the dried tea residues.

A WSC content higher than 5% is a basic condition for ensuring the quality of silage fermentation [36]. Nearly 80% of WSCs are consumed during the silage process since the LAB attached to the surface of silage material utilize WSCs to rapidly produce organic acids and inhibit other harmful microorganisms [44,45]. In this study, the WSC contents of raw alfalfa materials and other tea residues were less than 5%, except for B and W residues, which indicated that it would be challenging to sufficiently preserve alfalfa with tea residue as a silage additive. The WSC content in all the treatments decreased after 90 days of fermentation while there were large differences in the WSC content among the different tea residue treatments (Table 2 and Table 3), and the degree of WSC reduction in the B treatments was minimal. These differences might be caused by differences in the initial WSC content and microbial diversity of the different tea residues (Table 1).

Except for those in the W group, the CP contents in the other tea residue treatment groups were greater than those in the CK treatment group and fresh alfalfa before silage (*p* < 0.05), which could be explained by the greater CP content of the dried tea residue resulting from the different processing techniques (Table 1). In addition, the contents of NH_3_-N, FAA-N, and PN in the ensiled alfalfa decreased in the different tea residue treatments (*p* < 0.05), and the decreasing effect increased as the tea residue addition level increased. According to the study conducted by Ohshima et al. [46], the changes in protein composition in silage are mainly related to plant enzymes and microbial activity while the NH_3_-N content is mainly affected by the deamination induced by Clostridium proteolysis in silage. The NH_3_-N content is the main parameter commonly used to assess the protein hydrolysis of silage, When the NH_3_-N content of silage is less than 10% TN, it is considered one of the characteristics of silage preservation [39,47]. However, although the NH_3_-N content of each treatment in this study was lower than this NH_3_-N content, it could not be used as an indicator of good alfalfa silage fermentation. The reason for the decrease in NH_3_-N content in the present study may have been that the polyphenols in tea residue combine with proteins to form complexes, which makes it difficult for proteins to be degraded by microbial activity, which may also be one of the factors that reduce proteolysis [30]. Under the assumption that all the proteins contained in the added tea residues were true proteins, we calculated the loss of true proteins by converting them to nitrogen content and removing them. We found that after removing the protein nitrogen contained in the tea residues, the added tea residue treatments reduced the NPN content of alfalfa and the TPN content remained basically unchanged. The results indicated that tea residue addition could effectively preserve the TPN of alfalfa. This result is essentially consistent with the results of Wang et al. [48], and similar results were obtained in the study by He et al. [49]; the addition of gallic acid reduced the NH_3_-N content of silage and the abundance of *Cylindrium* spp.

Previous studies have shown that differences in CP content, metabolic energy, and NDF and ADF contents in silage could alter the IVDMD [50,51,52]. In this study, the IVDMD of alfalfa silage was above 65% whether or not tea residue was added, which was similar to the results of Xue et al. [53]. They also reported that the IVDMD of maize silage was 50%, which was lower than that of alfalfa silage; this might have been related to the high CP content and low fiber content of alfalfa, which are thought to be conducive to silage degradation. This study showed that the IDVMD and CP contents of the tea residue groups were greater than those of the CK group, which was consistent with the results of Du et al. [54], who also showed that the IVDMD was positively correlated with the CP content and negatively correlated with the NDF and ADF contents. Our results showed that the addition of tea residue to alfalfa silage improved the degradation rate of IVDMD.

The pH values of all the tea residue treatment groups in this study were lower than that of the CK, but the content of organic acids (LA and AA) was also significantly lower than that of the CK (*p* < 0.05). Vuong et al. [55] reported that after soaking green tea in water at a pH ranging from 4 to 8, the pH of green tea extract remained stable at approximately 5.3, which was attributed to the buffering capacity of green tea and its extracts. The inconsistency between the pH and volatile fatty acid contents of the tea residue treatments after ensiling may have been due to the buffering energy of the tea residues. Lower levels of organic acids (LA and AA) were detected in the tea residue treatments than in the CK group. Our research revealed that tea leaves can inhibit the activity of LAB, especially *Lactobacillus*, a fast acid-producing LAB, during the silage process, which influences the metabolic acid-producing ability of LAB [56]. Except in the D group, there were no significant differences in the PA contents between the other tea residue groups and the CK group (*p* < 0.05), possibly because the production of PA was not affected by the addition of tea residue.

Except for the D group, the DPPH and ABTS scavenging capacities and T-AOC in the B, G, W, and Z treatments of silage alfalfa were significantly greater than those in the CK treatment (Figure 2). Previous studies have shown that the DPPH and ABTS scavenging capacities and T-AOC are related to polyphenols and catechins in tea, which contain hydroxyl groups in their structure and thus can scavenge reactive oxygen species [57]. For these results in the study, that could be the polyphenols remain as phenolics after decomposition during silage fermentation while maintaining their antioxidant activity [26,58,59]. During the manufacturing of ripe Pu’er tea, humidification and fungal fermentation are needed, and these cause more damage to phenolic compounds and a significant loss in antioxidant potential [60,61].

The results of the 16S RNA sequence showed that the Chao 1, Ace, Shannon, and Simpson indices of all the treatments were lower than those of the CK treatment, except for the B5 and D5 treatments. Zheng et al. [62] reported that the bacterial diversity and richness of *Saccharum arundinaceum* decreased with chop treatment. It was speculated that adverse environmental conditions inhibited the growth of certain microorganisms [63], and in our study, the tea polyphenols contained in tea residue might have antibacterial effects, which affect the activity of bacteria in silages [56]. In addition, Firmicutes was the dominant phylum in each treatment after 90 days. However, its abundance varied, which may have been due to the different tea polyphenol concentrations in these tea residues resulting from the different processing methods [27]. Moreover, the abundance of Proteobacteria and Cyanobacteria increased with the addition of tea residue (Figure 3A). He et al. [49] reported that the abundances of Proteobacteria and Cyanobacteria were slightly greater in tannin-rich *Neolamarckia cadamba* leaf silage, and the addition of polyethylene glycol (a tannin inactivator) decreased the abundance of these bacteria, especially Cyanobacteria. This result indicated that tannins can decrease the abundance of Firmicutes and increase the abundance of Proteobacteria and Cyanobacteria in *N. cadamba* leaf silage, which was speculated to be due to alterations in the abundance of bacterial communities by tannins (polyphenolic substances). However, Huang reported the opposite results in alfalfa silage treated with tea polyphenols. In addition to tea polyphenols [27], other substances contained in tea residue might also have antibacterial effects, affecting the activity of bacteria during the ensiling process.

At the genus level, the abundances of bacteria differed among the five tea residue treatments, which was similar to the results for bacteria at the phylum level. This could have been because the addition of tea residue influences LAB growth, preventing the development of a dominant community in silage and inhibiting the growth of other miscellaneous bacteria [49]. The high abundance of other type bacteria may have been due to the relatively poor development of Cyanobacterial taxonomy wherein most Cyanobacteria cannot be cultured and characterized [27]. In addition, although the abundance of bacteria at the genus level was quite different in this study compared with other silage studies [43,64], the bacterial species detected in this study were basically similar to those in other studies. This suggests that the addition of tea grounds to alfalfa silage may only inhibit the growth of bacteria but not kill them or cause them to mutate. *Bacillus* was detected in the D treatment groups, and its relative abundance increased as the proportion of tea residue proportion increased. This finding is consistent with the findings of Xue et al. [65]. Another study explored the effects of microorganisms on the fermentation of Pu’er tea through metagenomics and metabolomics and revealed that the presence of *Bacillus* can also be detected after 30 days of natural solid-state fermentation of Pu’er tea [66]. Therefore, we speculated that the presence of Bacillus in the D treatment group might have resulted from the presence of D.

## 5. Conclusions

In summary, this study showed that the addition of dried tea residues can effectively inhibit the protein degradation of alfalfa silage, reduce the NPN fraction and AA content, and improve the antioxidant activity of alfalfa (DPPH, ABTS, and T-AOC). Among these treatments, the greatest effect was observed with the addition of G10. In addition, this study revealed that the effect of either dried tea residue on the nutritional quality of alfalfa silage was greater than the effect on fermentation quality. Based on the nutrient composition, the addition of black and green tea residues to alfalfa silage can improve its quality, and they have the potential to be used as silage additives.

## Figures and Tables

**Figure 1 microorganisms-12-00889-f001:**
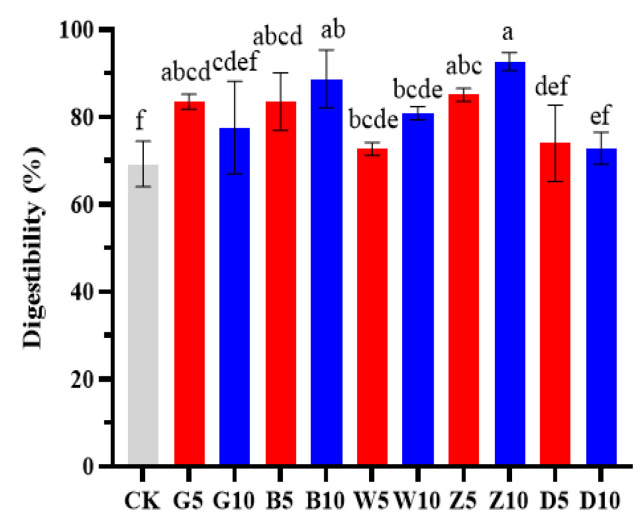
Effects of tea residues on in vitro the dry matter digestibility of alfalfa silage. Different lowercase letters represent significant differences between different treatments (*p* < 0.05). Abbreviations: CK, control; G, add green tea residue; B, add black tea residue; W, add white tea residue; Z, add Pu’er raw tea residue; D, add Pu’er ripe tea residue.

**Figure 2 microorganisms-12-00889-f002:**
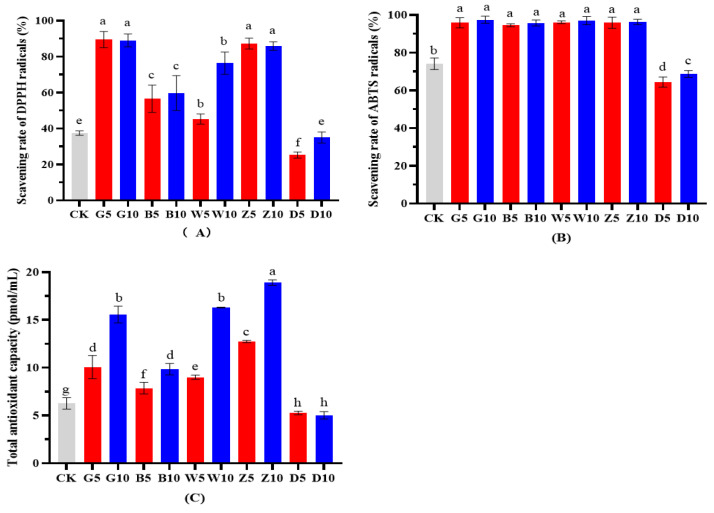
The effects of tea residues on the DPPH (**A**), ABTS (**B**) radical scavenging, T-AOC (**C**) of alfalfa silage. Different lowercase letters represent significant differences between different treatments (*p* < 0.05). Abbreviations: CK, control; G, add green tea residue; B, add black tea residue; W, add white tea residue; Z, add Pu’er raw tea residue; D, add Pu’er ripe tea residue.

**Figure 3 microorganisms-12-00889-f003:**
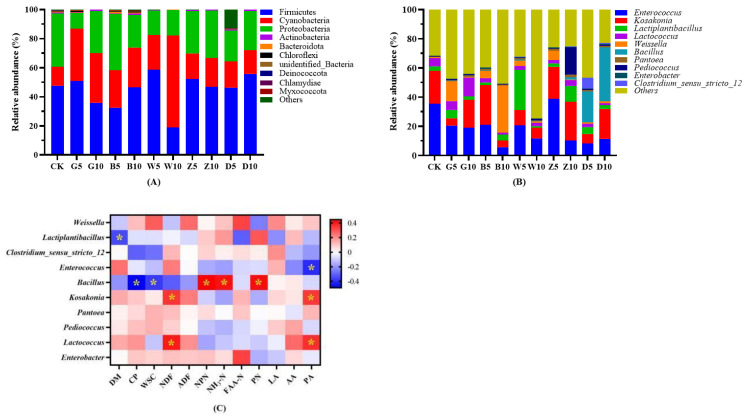
The bacterial community of the phylum (**A**) and genus (**B**) and Spearman correlation heatmap of the top tenth genus and silage fermentation properties (**C**). R is presented in different colors. The legend on the upper right side is a range of colors for different R values. * means *p* < 0.05.

**Table 1 microorganisms-12-00889-t001:** Nutritional compositions of materials before silage.

Items	Alfalfa	Green Tea Residues	Black Tea Residues	White Tea Residues	Pu’er Raw Tea Residues	Pu’er Ripe Tea Residues
DM (%FW)	41.6 ± 1.0	—	—	—	—	—
CP (%DM)	22.4 ± 1.0	33.8 ± 0.1	29.8 ± 0.1	14.5 ± 0.2	27.2 ± 0.1	30.9 ± 0.3
WSC (%DM)	4.0 ± 0.1	4.2 ± 0.6	7.8 ± 0.9	8.5 ± 0.1	4.5 ± 0.4	0.4 ± 0.1
NPN (%DM)	1.0 ± 0.1	1.45 ± 0.1	1.2 ± 0.1	0.65 ± 0.1	1.3 ± 0.1	1.0 ± 0.1
NDF (%DM)	31.8 ± 2.0	33.6 ± 1.0	33.8 ± 0.2	33.9 ± 0.5	26.0 ± 0.7	44.8 ± 0.2
ADF (%DM)	18.4 ± 2.4	8.13 ± 0.24	14.0 ± 0.6	18.6 ± 0.5	7.74 ± 0.63	36.8 ± 0.2
DPPH (%)	25.3 ± 2.2	91.5 ± 4.5	95.3 ± 0.9	94.1 ± 0.9	91.1 ± 0.4	43.4 ± 5.7
ABTS (%)	60.8 ± 2.8	92.5 ± 0.3	99.3 ± 0.4	90.4 ± 1.6	92.3 ± 0.7	91.4 ± 1.3
T-AOC (pmol/mL)	4.4 ± 0.2	18.8 ± 0.1	18.3 ± 0.4	17.6 ± 0.5	19.3 ± 0.4	4.5 ± 0.2

Abbreviations: FW, fresh weight; DM, dry matter; CP, crude protein; WSC, water-soluble carbohydrates; NPN, non-protein nitrogen; NDF, neutral detergent fiber; ADF, acid detergent fiber; DPPH, DPPH free radical scavenging; ABTS, ABTS free radical scavenging; T-AOC, total antioxidant capacity.

**Table 2 microorganisms-12-00889-t002:** Chemical properties and organic acids of alfalfa silage in the control group.

Items	Content
DM (%FW)	36.4 ± 0.4
CP (%DM)	21.5 ± 0.2
WSC (%DM)	1.2 ± 0.1
NDF (%DM)	21.6 ± 0.6
ADF (%DM)	10.5 ± 0.2
TPN (g/kg TN)	347.9 ± 6.3
NPN (g/kg TN)	652.0 ± 6.3
NH_3_-N (g/kg TN)	34.3 ± 12.5
FAA-N (g/kg TN)	90.8 ± 6.2
PN (g/kg TN)	526.8 ± 12.1
pH	6.0 ± 0.1
LA (%DM)	4.3 ± 0.2
AA (%DM)	2.8 ± 0.5
PA (%DM)	1.5 ± 0.1
BA (%DM)	ND

Abbreviations: CP, crude protein; WSC, water-soluble carbohydrates; NDF, neutral detergent fiber; ADF, acid detergent fiber; TPN, true-protein nitrogen; NPN, non-protein nitrogen; NH_3_-N, ammonia nitrogen; FAA-N, free amino acid nitrogen; PN, peptide nitrogen; LA, lactic acid; AA, acetic acid; PA, propionic acid; BA, butyric acid; ND, not detected.

**Table 3 microorganisms-12-00889-t003:** Effects of tea residues on the nutritional compositions of alfalfa silage.

Treatments		DM (%FW)	CP (%DM)	WSC (%DM)	NDF (%DM)	ADF (%DM)
Types of Tea Residues	Proportions (%FW)					
G	5	41.15 b	23.82 de	1.10 d	17.22 d	8.64 e
	10	43.59 a	25.22 b	1.31 c	24.34 a	14.63 b
B	5	39.80 c	23.32 e	1.76 a	20.91 c	16.55 a
	10	42.99 a	24.34 cd	1.57 b	22.60 b	12.67 c
W	5	38.82 cd	21.41 f	1.08 d	21.36 bc	12.23 c
	10	41.17 b	21.06 f	1.37 c	20.36 c	11.60 c
Z	5	38.08 d	23.27 e	0.87 e	17.75 d	10.20 d
	10	41.19 b	24.76 bc	0.81 e	21.16 c	12.01 c
D	5	38.91 cd	24.23 cd	0.60 f	25.36 a	16.74 a
	10	40.95 b	25.85 a	0.79 e	25.01 a	16.95 a
SEM		0.688	0.407	0.056	0.907	0.833
*p*-value	T	***	***	***	***	***
	P	***	***	***	***	*
	T × P	ns	***	***	***	***

Different lowercase letters in the same column represent significant differences (*p* < 0.05). Abbreviations: G, add green tea residue; B, add black tea residue; W, add white tea residue; Z, add Pu’er raw tea residue; D, add Pu’er ripe tea residue; CP, crude protein; WSC, water-soluble carbohydrates; NDF, neutral detergent fiber; ADF, acid detergent fiber. SEM, standard error of the mean; T, types of tea residue; P, proportions of tea residue added; T × P, The interaction between types of tea residue and proportions of tea residue added; significance levels: ***, *p* < 0.001; *, *p* < 0.05; ns, not significant.

**Table 4 microorganisms-12-00889-t004:** Effects of tea residues on the nitrogen compositions of alfalfa silage.

Treatments		TPN (g/kg TN)	NPN (g/kg TN)	NH_3_-N (g/kg TN)	FAA-N (g/kg TN)	PN (g/kg TN)
Types of Tea Residues	Proportions (%FW)					
G	5	420.8 c	579.2 b	26.0 abcd	85.3 abc	467.9 ab
	10	469.4 b	530.6 c	18.6 de	83.0 abc	429.0 cd
B	5	420.5 c	579.5 b	23.0 bcde	106.2 a	450.3 bc
	10	458.6 b	541.4 c	17.1 e	71.5 bc	452.8 bc
W	5	394.4 d	605.6 a	32.9 a	79.1 bc	493.6 a
	10	431.7 c	568.3 b	23.9 bcde	75.6 bc	468.8 ab
Z	5	425.4 c	574.6 b	27.6 abc	64.4 bc	482.6 a
	10	467.2 b	532.8 c	20.4 cde	67.7 bc	444.7 bc
D	5	423.3 c	576.7 b	30.8 ab	75.6 bc	470.3 ab
	10	507.4 a	492.6 d	29.8 ab	59.4 c	403.4 d
SEM		7.634	10.469	5.054	16.099	18.265
*p*-value	T	***	***	***	*	***
	P	***	***	***	*	***
	T × P	**	***	ns	ns	*

Different lowercase letters in the same column represent significant differences (*p* < 0.05). Abbreviations: G, add green tea residue; B, add black tea residue; W, add white tea residue; Z, add Pu’er raw tea residue; D, add Pu’er ripe tea residue; TPN, true-protein nitrogen; NPN, non-protein nitrogen; NH_3_-N, ammonia nitrogen; FAA-N, free amino acid nitrogen; PN, peptide nitrogen. SEM, standard error of the mean; T, type of tea residue; P, proportion of tea residue added; T × P, the interaction between types of tea residue and proportion of tea residue added. Significance level: ***, *p* < 0.001; **, *p* < 0.01; *, *p* < 0.05; ns, not significant.

**Table 5 microorganisms-12-00889-t005:** Effects of tea residues on the fermentation quality of alfalfa silage.

Treatments		pH Value	LA (%DM)	AA (%DM)	PA (%DM)	BA (%DM)
Types of Tea Residues	Proportions (%FW)					
G	5	5.78 c	1.72 f	1.00 c	1.43 a	ND
	10	5.81 bc	2.13 de	1.60 ab	1.54 a	ND
B	5	5.91 abc	3.60 a	1.61 ab	1.46 a	ND
	10	5.71 c	3.48 ab	1.63 ab	1.52 a	ND
W	5	5.86 abc	2.39 d	1.61 ab	1.48 a	ND
	10	5.80 bc	3.23 b	1.24 bc	1.29 ab	ND
Z	5	5.90 abc	1.90 ef	1.84 a	1.27 ab	ND
	10	5.73 c	2.21 de	1.85 a	1.43 a	ND
D	5	6.07 a	3.17 bc	1.37 bc	1.16 b	0.35
	10	5.91 abc	2.87 c	1.12 c	1.08 b	0.45
SEM		0.098	0.156	0.145	0.170	—
*p*-value	T	*	***	***	***	—
	P	*	**	ns	ns	—
	T × P	ns	***	***	ns	—

Different lowercase letters in the same column represent significant differences (*p* < 0.05). Abbreviations: G, add green tea residue; B, add black tea residue; W, add white tea residue; Z, add Pu’er raw tea residue; D, add Pu’er ripe tea residue; LA, lactic acid; AA, acetic acid; PA, propionic acid; BA, butyric acid; ND, not detected. SEM, standard error of the mean; significance level: ***, *p* < 0.001; **, *p* < 0.01; *, *p* < 0.05; ns, not significant.

**Table 6 microorganisms-12-00889-t006:** Alpha diversity of bacterial of alfalfa silage after adding dried tea residues.

Treatments		OTUs	Shannon	Simpson	Chao1	Ace	Coverage
Types of Tea Residue	Proportions (%FW)						
CK	—	385	3.33	0.75	444	448	0.99
G	5	293	3.15	0.74	336	353	0.99
	10	187	2.60	0.66	203	218	0.99
B	5	413	3.53	0.80	498	495	0.99
	10	317	2.87	0.59	368	370	0.99
W	5	274	2.91	0.67	327	344	0.99
	10	183	2.41	0.55	204	216	0.99
Z	5	297	3.21	0.77	336	352	0.99
	10	171	3.20	0.80	202	208	0.99
D	5	315	3.75	0.79	429	367	0.99
	10	198	2.90	0.75	235	242	0.99

Abbreviations: CK, control, G, add green tea residue; B, add black tea residue; W, add white tea residue; Z, add Pu’er raw tea residue; D, add Pu’er ripe tea residue.

## Data Availability

Data are contained within the article.

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
