# Peer review of "Effects of Dried Tea Residues of Different Processing Techniques on the Nutritional Parameters, Fermentation Quality, and Bacterial Structure of Silaged Alfalfa"

_microorganisms, 2024, doi:10.3390/microorganisms12050889_

Round 1

Reviewer 1 Report

Comments and Suggestions for Authors

Line 38: Verify the bolded parts

Line 49: verify typos here and through the text.

Lines 51-54: Please, include references.

Lines 68-71: Please, include numbers! What is estimated for these amounts for the country? Or for local production? Or for the region? Or for the world?

Line 87: ????????!!!!

Line 89-91: Please include some references

Line 123: verify the pontuaction here and through the text.

Sections 2.2 and 2.3 (especially 2.3.2): Please, revise carefully!

Tables 1 and 2: Standardize the number of decimals

Line 249: Please verify the space between words here and through the text

Lines 354-   : Please, verify the italic for scientific names

References: verify bolded words

The authors have prepared a paper with merit and novelty. However, they should revise all the manuscript carefully before a possible acceptance.

Author Response

We are very grateful to the experts for your valuable comments on our manuscript, and we have completed the revision of the manuscript based on the revisions you have given. Please see the attachment. Please see the attachment.

Reviewer 2 Report

Comments and Suggestions for Authors

The title does not read so well, consider changing from "Effect of Dried Tea Residues of Different Processing Techniques on the Nutritional, Fermentation Quality and Bacterial Structure of Silage Alfalfa" to "Effect of Dried Tea Residues of Different Processing Techniques on Nutritional Parameters, Fermentation Quality and Bacterial Structure of Silage Alfalfa"

Line 12 processing byproduct produced in tea production --> 3 x word repetition, rephrase

Line 44 will remain the largest producer of tea in 2020--> was the largest producer of tea in 2020

line 87: wsrf4byhrgvecftghfgvrftfgvgvgvtgvgvor ??

line 100: on September 25, 2021: Why do you report the results 2 1/2 years later?

Why did the white tea residue not behave similarly to green tea residue?

Are there regional differences from tea residue, e.g. impact of soil?

Comments on the Quality of English Language

minor editing only

Author Response

(The authors gave the same response as above.)

Round 2

Reviewer 1 Report

Comments and Suggestions for Authors

After the revisions the manuscript is ready to be accepted